# The Dynamic of Polyphenols Concentrations in Organic and Conventional Sour Cherry Fruits: Results of a 4-Year Field Study

**DOI:** 10.3390/molecules25163729

**Published:** 2020-08-15

**Authors:** Agnieszka Głowacka, Elżbieta Rozpara, Ewelina Hallmann

**Affiliations:** 1Research Institute of Horticulture, Konstytucji 3 Maja 1, 96-100 Skierniewice, Poland; agnieszka.glowacka@inhort.pl (A.G.); elzbieta.rozpara@inhort.pl (E.R.); 2Department of Functional and Organic Food, Institute of Human Nutrition Sciences, Warsaw University of Life Sciences, Nowoursynowska 159c, 02-776 Warsaw, Poland

**Keywords:** *Prunus cerasus*, organic farming, conventional farming, polyphenols, anthocyanins

## Abstract

Sour cherry fruits are a perfect source of polyphenols, including flavonols, phenolic acids and anthocyanins. According to the literature, organic fruits contain more bioactive compounds, especially polyphenols, compared to conventional fruits. Given that only one two-year experiment on the status of polyphenols in sour cherry fruits in an organic farm was previously conducted and found in the existing scientific literature, the aim of this study was to analyse and compare the concentration of bioactive compounds in organic and conventional sour cherries and to determine the effects of cultivation year and the proper cultivar. Four sour cherry cultivars (“Oblacińska”, “Kelleris 16”, “Pandy 103” and “Dobroceni Bötermo”) harvested in organic and conventional experimental orchards were assessed in this study. The dry matter and polyphenol contents in the fruits were determined. We observed a significantly higher concentration of dry matter in organic samples only in 2015 and 2017. In the case of total polyphenols, including phenolic acid (2015 and 2017–2018), and total flavonoids, including quercetin-3-*O*-rutinoside, the higher concentration was found in 2016 and 2018. Two individual anthocyanins were identified in sour cherry fruits. Cyanidin-3-*O*-rutinoside is the predominant form in the pool of total anthocyanins.

## 1. Introduction

Sour cherries are one of the most popular stone fruits in Europe. According to the Eurostat database, Poland was the largest conventional sour cherry producer in Europe in 2018 (27.97 thousand ha and 200.63 thousand tonnes) followed by Turkey (22.00 thousand ha and 184.00 thousand tonnes) and Serbia (17.57 thousand ha and 128 thousand tonnes). Organic sour cherry production levels are much smaller, but Poland remains one of the largest producers. Sour cherries are one of the most important stone fruits produced by Polish farms. In 2018, Poland had 1.06 ha of organic sour cherry orchards that produced 707 tonnes (Eurostat database). Sour cherry fruits are consumed in the fresh stage as a dessert fruit, and they are an important material for processing frozen cherry fruits, juices, and jams [1,2,3,4]. Modern technologies are producing new products, such as capsules with freeze-dried sour cherry pomace [5]. Organic farming is a very strict cultivation system, supported by law. In this system, only natural kinds of fertilizers are allowed (green manure, animal manure, and compost) and natural plant protection methods (natural predators, pheromone traps, and yellow sticky boards). Sour cherries are good sources of bioactive compounds in the human diet. The bioactive compounds mainly include polyphenols, such as flavonols, phenolic acids, and anthocyanins [6,7,8]. Plants produce polyphenols in response to biotic and abiotic stress [9]. A high concentration of phenolic compounds has a positive effect on the human body. The aetiologies of many metabolic diseases involve excessive free radical production. Polyphenolic compounds are well known as free radical scavengers. These compounds may be useful in the prevention of the development of many chronic diseases, including various cancers, obesity, and cardiovascular diseases [10,11,12].

Only a limited number of experiments on organic sour cherries have been reported, and these studies exclusively focused on economic aspects and yields [13,14,15,16]. Information on organic vs. conventional fruit quality and the polyphenol composition of fruits is limited [17], and results from long-term experiments are especially lacking. The aim of the present work is to assess the effect of organic vs. conventional farm management practices on polyphenol levels in four sour cherry cultivars during long-term cultivation.

## 2. Results and Discussion

### 2.1. Dry Matter Content

The organic sour cherry contained significantly more dry matter but only in the first (*p* = 0.0078) and second (*p* = 0.011) experimental years. In 2017, we observed the highest and significant amount of dry matter in conventional fruits (*p* = 0.0087); however, in 2018, no differences were noted between farming systems (Table 1, Table 2, Table 3 and Table 4). Some authors demonstrated that fruits produced under organic conditions contained more dry matter [18,19,20]. On the other hand, some experiments reported contrary findings [21]. The increased level of dry matter content in organic fruits can be explained by the “water swelling” phenomenon which is characteristic of conventional fruits. Trees in conventional orchards intake mineral fertilizers dissolved in water. Soft tissues of plants grown under conventional methods collect more water compared with those grown under organic conditions [22].

Among the examined cultivars, “Oblacińska” cv. exhibited the highest dry matter content. In three out of four experimental years, “Oblacinska” cv. fruits contained significantly more dry matter (*p* = 0.002; *p* = 0.0037; *p* < 0.0001) compared with the other experimental sour cherry cultivars. This was observed in the organic production system especially (Table 1, Table 2, Table 3 and Table 4). Konopacka et al. (2014) presented similar results [23]. “Oblacińska” cv. contained significantly more dry matter compared with “Morina” and “Koral” cvs.

### 2.2. Total Polyphenols—Phenolic Acids Content

In the following years of the experiment (2016–2018), the total content of polyphenols was significantly higher in organic cherry fruits. In the first experimental year (2015), trends regarding high concentrations of total polyphenols were only observed some cultivars of sour cherries grown via organic and conventional methods. However, in the following three years, the level of total polyphenols was stable and always significantly increased under organic cultivation conditions (Table 1, Table 2, Table 3 and Table 4). Nagy-Gasztonyi et al. (2010) reported similar observations [17]. In the first year of the experiment, the level of total polyphenols was 65.79 mg/100 g FW in organic sour cherries and 60.74 mg/100 g FW in sour cherries grown under conventional methods. In the second year, these levels were considerably increased to 94.37 mg/100 g FW for organic sour cherries and 81.44 mg/100 g FW for sour cherries grown under conventional methods. Among the examined cultivars, “Keleris 16” cv. exhibited the highest levels of total polyphenols in both cultivation systems (organic and conventional) in 2015, 2017 and 2018 (Table 1, Table 2, Table 3 and Table 4). Plants produce phenolic compounds, especially phenolic acids, in response to biotic and abiotic stressors. Phenolic acids are well known as “natural pesticides” [24,25]. The higher concentrations of phenolic acids in plants from organic farm management practices could be an effect of the lack of pesticide use. As previously noted, the use of chemical plant protection is completely forbidden in organic orchards. On the other hand, pest and diseases are a common problem in organic and conventional farming systems. Thus, organic plants are protected by the production of high levels of phenolic acids in tissues. Among individual phenolic acids, two compounds that belong to this group were identified: ferulic and chlorogenic acids. Organic sour cherries contained significantly higher levels of ferulic acid throughout the course of the experiment (Table 1, Table 2, Table 3 and Table 4). We did not observe any trends regarding ferulic acid content in individual cultivars. Chlorogenic acid is one of the most important phenolic acids in sour cherry fruits. Organic sour cherries contained significantly more chlorogenic acid in 2015, 2017 and 2018 (Table 1, Table 2, Table 3 and Table 4). The cultivar “Kelleris 16” contained the highest chlorogenic acid content. As was pointed in the experiment with sweet cherry fruits, both chlorogenic and ferulic acids play an important role as antifungal agents [26]. It is worthwhile to point out that in every year of the presented experiment, even up to four chemical fungicides for plant protection against different fugues were used in conventional orchard. Trees in the organic farming system were protected only by one authorized protection agent—copper. In such a situation, plants in the organic system were forced to synthesize a higher level of chlorogenic and ferulic acids used by plants as antifungal compounds. Wojdyło et al. (2014) reported similar findings [8].

### 2.3. Flavonoids Content

In the present experiment, organic sour cherries contained significantly more total flavonoids (*p* < 0.0001 and *p* = 0.0014 in 2016 and 2018, respectively). The total flavonoid content increased in sour cherry fruits yearly to reach the highest concentration in the last experimental year (2018). Nagy-Gasztonyi et al. (2010) reported similar findings [17]. During the first three years of the experiment, a decrease in the total flavonoids content in cherry fruits was observed year by year. Only in the fourth year, the concentration of total flavonoids increased to the highest level. In 2015 and 2017, “Oblacińska” cv. had the highest total flavonoid content in both organic and conventional systems, whereas “Keleris 16” cv. exhibited the highest levels in 2018 (Table 1, Table 2, Table 3 and Table 4). Significant variability was found in total flavonoid content among “Oblacińska” cv. clones with values ranging from 6.36 to 23.46 mg/100 g FW [27]. Four flavonol compounds were identified: two quercetin derivatives, kaempferol-3-*O*-glucoside, and myricetin (Table 1, Table 2, Table 3 and Table 4). The obtained results are inconsistent with those reported by Wojdyło et al. (2014) [8]. The main flavonol present in sour cherry fruits is quercetin-3-*O*-rutinoside. In our experiment, organic sour cherry fruits consistently contained significantly more flavonol compounds (*p* < 0.0001, *p* = 0.0011, *p* < 0.0001 and *p* < 0.0001 in 2015–2018, respectively). Organic sour cherries contained 2.64 mg/100 g FW and 2.72 mg/100 g FW in 2008 and 2009, respectively, whereas conventional sour cherries contained 2.45 mg/100 g FW and 2.69 mg/100 g FW, respectively [17]. In a 4-year experiment, we observed that in 2015 and 2017, the highest concentration of quercetin-3-*O*-rutinoside was found in the organic “Keleris 16” cv. fruits and the conventional “Oblacińska” cv. fruits. In 2016 and 2018, on the other hand, they were organic fruits “Debreceni Botermo” cv. and conventional fruits “Keleris 16” cv. The higher temperature in time of fruits collecting could have an impact on quercetin-3-*O*-rutinoside concentration in sour cherry fruits. Similar findings were presented with cornelian cherry fruits collected in different regions of Bośnia and Hercegovina. Fruits collected from the places with the highest average 24-hours temperature were characterized by a higher content of quercetin-3-*O*-rutiniside [28]. In our experiment, in 2017, we observed the highest concentration of quercetin-3-*O*-rutinoside in sour cherry fruits. In this year the average 24-hour temperature was the highest compare to the rest of the years in the experiment (Appendix A). Kaempferol-3-*O*-glucoside was identified in conventional sour cherry fruits in significantly higher concentrations compared to organic ones (*p* = 0.0031 in 2015; *p* < 0.0001 in 2017 and *p* < 0.0001 in 2018). A similar result was observed in the experiment with organic and conventional herbs, strawberries and potatoes [29,30,31].

### 2.4. Anthocyanins Content

Anthocyanins are one of the most important bioactive compounds in sour cherry fruits. The organic sour cherry only contained significantly more total anthocyanins in 2016 and 2018 (*p* < 0.0001 and 0.0015, respectively). Some experiments showed that organic fruits contained significantly more total anthocyanins [21,22,23,24,25,26,27,28,29,30,32]. Total anthocyanins were present at the highest levels in “Oblacińska” cv. (94.08 mg/100 g FW in 2015 and 84.06 mg/100 g FW in 2017), in both organic and conventional systems. In 2018, the highest was cultivar “Keleris 16”. “Oblacińska” cv. contained 35.77 mg/100 g FW total anthocyanins [6]. In the present experiment, two anthocyanins were identified, and both are cyanidin derivatives: cyanidin-3-*O*-rutinoside and cyanidin-3-*O*-glucoside. Sour cherry fruits contained three-fold more quercetin-3-*O*-rutinoside compared to sour cherries grown under conventional methods. Among the examined cultivars, “Oblacińska” cv. exhibited the highest cyanidin-3-*O*-rutinoside levels at 87.38 mg/100 g FW and 78.86 mg/100 g FW in 2015 and 2017, respectively (Table 1, Table 2, Table 3 and Table 4). Contrary results for “Oblcińska” cv. were presented by Šimunić et al. (2005) [33]. The second identified anthocyanin is cyanidin-3-*O*-glucoside. Cyanidin-3-*O*-glucoside contents in sour cherry cultivars range from 0.88 mg/100 g FW to 1.31 mg/100 g FW [34]. These anthocyanidin glucosides are characteristic compounds found in cherry fruits. Therefore, some differences in cyanidin-3-*O*-glucoside levels in cherry fruits are reported in the literature [35]. In the present experiment, we observed a high variation in cyanidin-3-*O*-glucoside levels among the examined cultivars in all experimental years, with values ranging from 3.26 mg/100 g FW to 6.70 mg/100 g FW (Table 1, Table 2, Table 3 and Table 4).

Principal component analysis (PCA) showed a high and significant overall variation of 78.56% which was explained by PC1 and PC2 (Figure 1A). The degree of dependence between the farm management systems and the factors of dry matter (DM), total polyphenols (TP), total phenolic acids (TPA), ferulic acid (FA), chlorogenic acid (ChLA), total flavonoids (TF), total flavonols (TFl), quercetin-3-*O*-rutinoside (Q-3-*O*-R), quercetin-3-*O*-glucoside (Q-3-*O*-G), total anthocyanins (TA), cyanidin-3-*O*-rutinoside (Cy-3-*O*-R), and cyanidin-3-*O*-glucoside (Cy-3-*O*-G) contents were particularly important. Of note, organic and conventional sour cherry samples were located in different parts of the graph, revealing diversity between samples from different farm management systems. Figure 1B shows a significant overall variation of 67.74% which was explained by PC1 and PC2. A degree of dependence was noted between the examined sour cherry cultivars and the indicated factors. The examined cherry cultivars can be grouped into individual parts of the graph. Cultivars “Kelleris 16” and “Oblacińska” are primarily dependent on total polyphenols (TP), total phenolic acids (TPA), total anthocyanins (TA), quercetin-3-*O*-glucoside (Q-3-*O*-G) and kaempferol-3-*O*-glucoside (K-3-*O*-G).

## 3. Material and Methods

### 3.1. Chemicals

Acetonitrile (Sigma-Aldrich, Poland), acetic acid (99.9% purity, Chempur, Poland), deionized water (Sigma-Aldrich, Warsaw, Poland), methanol (Merck, Warsaw, Poland), ortho-phosphoric acid (Chempur, Warsaw, Poland), and phenolic compound standards, including ferulic acid (CAS 537-98-4), chlorogenic acid (CAS 327-97-9), kaempferol-3-*O*-glucoside (CAS 480-10-4), quercetin-3-*O*-glucoside (CAS 482-35-9), quercetin-3-*O*-rutinoside (CAS 207671-50-9), myrycetin (CAS 529-44-2), cyanidin-3-*O*-glucoside (CAS: 7084-24-4), and cyjanidin-3-*O*-rutinoside (CAS 18719-76-1), were used in this study. (Sigma-Aldrich, Supelco, Poland).

### 3.2. Plant and Fruit Origins

Experiments were performed during the period 2015–2018. Four sour cherry cultivars, including “Keleris 16”, “Olbacińska”, “Pandy 103”, and “Debreceni Bötermo”, were used. The organic orchard was located in Skierniewice (Nowy Dwór-Parcela; 51°52′ 0″ N 20°15′43″E), and the conventional orchard was located in Skierniewice (Dąbrowice; 52°18′56″ N 19°5′4″E). The distance between orchards was 16 km. All data on cultivation conditions are presented in Appendix A. In experimental organic orchards, trees were grafted onto *Prunus mahaleb* seedlings, were planted at a spacing of 4.5 × 2.5 m, in four replications, with three trees per plot. For the first two years, the soil in the orchard was kept in mechanical fallow. From the third year onward, mechanical fallow was still maintained in the rows of trees, while the inter-rows were kept under self-seeding grass cover. The tree crowns were trained in the form of a spindle. Light sanitary and rejuvenation pruning was carried out every year. From 2015, the sour cherry trees were watered via a drip irrigation system. From 2015, using the meteorological station located in the orchard, data were collected on weather conditions in order to assess their impact on the health status and yielding of the trees, and on fruit quality. The trees were managed in accordance with the principles of organic horticulture. In conventional orchard in the first two years, mechanical fallow was maintained in the orchard. In the third year, grass cover was introduced in the inter rows, which was mowed several times during the growing season, while herbicide fallow was maintained in the rows. The trees were irrigated via a drip system, and the protection and agrotechnical treatments were performed in accordance with the recommendations for orchards with conventional production.

All weather conditions at the time of sour cherry cultivation in 2015–2018 are presented in Appendix A.

### 3.3. Fruits Collecting and Sample Preparation

In every experimental year, fruits were collected at a similar time: 7 July 2015; 8 July 2016; 7 July 2017 and 9 July 2018. For fruit collecting, 3 of 12 trees from each cultivar were chosen. From one tree, 2 kg of fully ripens fruits were obtained. The scheme of fruits collecting was representative for combination and cultivars. We point attention to collect fruits from every place on the tree (up, down, inside and outside part of the crown). After collecting, samples were transported from orchards to the laboratory in cooling polystyrene boxes. In the laboratory, fruits were cut and pitted and prepared to dry matter measurement as well as freeze-dried. Each sample was divided into two parts. The first part was used for dry matter evaluation, and the second part was freeze-dried using a Labconco (2.5) freeze-dryer (Warsaw, Poland, −40 °C, pressure 0.100 mBa). After freeze-drying, the plant material was ground in a laboratory mill (A-11). The ground samples were then stored at -80 °C until the end of the analysis in every experimental year.

### 3.4. Dry Matter Measurements

The dry matter of sour cherry fruits was measured using the scale method before samples were freeze-dried. Empty glass beakers were weighed, filled with fresh cherries and weighed again. The samples were placed in a FP-25W Farma Play (Poland) dryer set to 105 °C for 72 h. After 3 days, the samples were cooled to 21 °C and weighed again. The dry matter content was calculated for the sour cherry samples based on their mass differences and reported as g/100 g FW [36].

### 3.5. Polyphenol Extraction and Identification

The polyphenol content in sour cherry fruit samples was analysed using the HPLC method [37]. Briefly, 100 mg of freeze-dried sample was weighed in a plastic tube, and 5 mL of 80% methanol was added. Samples were mixed using a vortex and incubated at 30 °C for 10 min in an ultrasonic bath. Next, the samples were centrifuged (first centrifugation at 6000 rpm and second centrifugation at 12,000 rpm). Then, 1 mL of supernatant was used for further HPLC (High-Performance Liquid Chromatography) analysis, injection volume 100 μL. The HPLC setup included two pumps (LC-20AD), autosampler (SIL-20AC), controller unit (CMB-20A), UV-Vis detector (SPD-20AV), column oven (CTO-20AC) (Shimadzu, USA Manufacturing Inc., Canby, OR, USA), and Phenomenex column Fusion-RP 80 Å (250 × 4.60 mm). Phenolic compound elution was performed using gradient flow with two mobile phases: (A) 10% acetonitrile and (B) 55% acetonitrile (pH 3.0 obtained using *ortho*-phosphoric acid). The programme was as follows: 0–21 min (95% Solvent A and 5% Solvent B), 22–25 min (50% Solvent A and 50% Solvent B), 26–27 min (20% Solvent A and 80% Solvent B), 28–32 min (20% Solvent A and 80% Solvent B), and 32–36 min (95% Solvent A and 5% Solvent B). The analysis time was 38 min with a flow rate of 1 mL/min and a wavelength range of 250–370 nm. Fluka and Sigma-Aldrich external standards were used for polyphenol identification.

### 3.6. Anthocyanin Extraction and Identification

Anthocyanin content was analysed using the HPLC method [37]. The first step of anthocyanin extraction (with 80% of methanol) was described step-by-step for polyphenols (see description Section 2.4). Next, 2.5 mL of supernatant was placed into a new plastic tube, and 2.5 mL of 10 M HCl and 5.0 mL 100% methanol were added. The samples were cooled and stored at 5 °C for 10 min. Briefly, 100 μL supernatant was used in HPLC analysis. Anthocyanins were eluted using an isocratic flow with one mobile phase (5% acetic acid, acetonitrile, and methanol (70:10:20) *v/v/v*). The flow rate was 1.5 mL/min, and the wavelength for detection was 530 nm. Anthocyanins were identified based on Sigma-Aldrich, external standards: cyanidin-3-*O*-rutinoside and cyanidin-3-*O*-glucoside.

### 3.7. Statistical Analysis

The chemical analysis results were statistically analysed using Statgraphics Centurion 15.2.11.0 software (StatPoint Technologies, Inc., Warrenton, VA, USA). The values presented in the tables are expressed as the mean values for organic and conventional farming systems (*n* = 12) and the four examined sour cherry cultivars (“Oblacińska”, “Kelleris 16”, “Pandy 103” and “Debreceni Bötermo”) (plot replication *n* = 3). The number of HPLC replication (laboratory replication *n* = 3). Statistical calculations were based on two-way analysis of variance using Tukey’s test (*p* = 0.05). The same letters indicate the lack of statistically significant differences between the examined groups. Standard error (SE) is provided for each mean value reported in the tables. Principal component analysis (PCA) was performed using XLSTAT Software (XLSTAT, 2020, NewYork, NY, USA) to categorize the farming systems used in 2015–2018 and individual cultivars examined in 2015–2018 based on their bioactive compound contents.

## 4. Conclusions

The present study confirms that organic farming systems may have a significant impact on the phenolic composition of sour cherry fruits. On the other hand, long-term experiments provide more specific and accurate answers to questions about the impact of the cultivation system on the phenolic status of cherry cultivars, if short-year experiments do not show clear results. It is worth pointing out that the vast majority of the research presented in the literature on the quality of organic fruits has exclusively focused on one-year experiments. Major changes in the chemical composition of sour cherry fruits are noted based on time and production methods in organic orchards. The choice of the proper cultivar has a significant impact on the polyphenol content of sour cherry fruits.

## Figures and Tables

**Figure 1 molecules-25-03729-f001:**
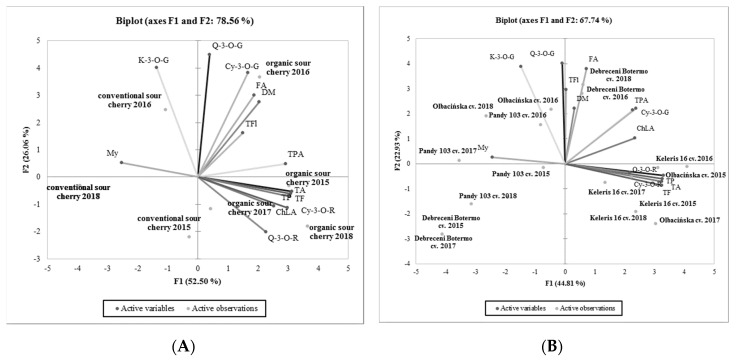
Principal component analysis (PCA) analysis showing the relationship between the polyphenols composition and (**A**) cultivation systems as well examined (**B**) cultivars of sour cherry in 2015–2018. Dry matter (DM,) total polyphenols (TP), total phenolic acids (TPA), ferulic acid (FA), chlorogenic acid (ChLA), total flavonoids (TF), total flavonols (TFl), quercetin-3-*O*-rutinoside (Q-3-*O*-R), quercetin-3-*O*-glucoside (Q-3-*O*-G), kaempferol-3-*O*-glucoside (K-3-*O*-G), myrycetin (My), total anthocyanins (TA), cyanidin-3-*O*-rutinoside (Cy-3-*O*-R), cyanidin-3-*O*-glucoside (Cy-3-*O*-G).

**Table 1 molecules-25-03729-t001:** The content of dry matter (in g/100 g FW) and polyphenols (in mg/100 g FW) in examined sour cherry cultivars in 2015.

Bioactive Compounds/Experimental Combination	Organic Sour Cherry (*n* = 12)					Conventional Sour Cherry (*n* = 12)					*p*-Value		
	Keleris 16 cv. (*n* = 3)	Olbacińska cv. (*n* = 3)	Pandy 103 cv. (*n* = 3)	Debreceni Botermo cv. (*n* = 3)	Mean for Organic	Keleris 16 cv. (*n* = 12)	Olbacińska cv. (*n* = 12)	Pandy 103 cv. (*n* = 12)	Debreceni Botermo cv. (*n* = 12)	Mean for Conventional	System (S)	Cultivar (C)	Interaction (SxC)
dry matter	14.9 ± 0.2 ^c^	17.6 ± 0.2 ^a^	15.5 ± 0.4 ^b^	16.0 ± 0.3 ^b^	16.0 ± 0.2 A	13.8 ± 0.4 ^d^	15.4 ± 0.5 ^b^	15.2 ± 0.8 ^b^	14.9 ± 0.4 ^c^	14.8 ± 0.3 B	0.0078	0.0092	<0.0001
total polyphenols	87.2 ± 1.2 ^b^	119.9 ± 1.2 ^a^	44.3 ± 1.4 ^e^	44.9 ± 9.2 ^e^	74.1 ± 0.7 A	74.7 ± 1.9 ^c^	84.6 ± 3.1 ^b^	69.2 ± 4.1 ^d^	62.6 ± 0.9 ^d^	72.8 ± 2.7 A	NS	<0.0001	0.0036
total phenolic acids	6.1 ± 0.1 ^a^	5.8 ± 0.1 ^b^	5.6 ± 0.2 ^b^	3.7 ± 0.1 ^c^	5.3 ± 0.0 A	5.3 ± 0.1 ^b^	4.0 ± 0.1 ^c^	3.6 ± 0.1 ^c^	2.5 ± 0.1 ^d^	3.9 ± 0.3 B	<0.0001	<0.0001	<0.0001
ferulic acid	1.2 ± 0.0 ^a^	1.0 ± 0.0 ^a^	1.1 ± 0.1 ^a^	0.8 ± 0.2 ^b^	1.0 ± 0.0 A	0.7 ± 0.0 ^b^	0.8 ± 0.0 ^b^	1.0 ± 0.0 ^a^	1.1 ± 0.0 ^a^	0.9 ± 0.0 A	0.0036	NS	<0.0001
chlorogenic acid	4.9 ± 0.0 ^a^	4.8 ± 0.1 ^a^	4.5 ± 0.1 ^a^	2.9 ± 9.0 ^c^	4.3 ± 0.0 A	4.6 ± 0.1 ^a^	3.2 ± 0.1 ^b^	2.7 ± 0.1 ^c^	1.4 ± 0.0 ^d^	3.0 ± 0.3 B	<0.0001	<0.0001	<0.0001
total flavonoids	81.1 ± 1.2 ^b^	114.1 ± 1.1 ^a^	38.6 ± 1.2 ^d^	41.3 ± 0.1 ^d^	68.8 ± 0.7 A	69.4 ± 1.8 ^c^	80.6 ± 3.0 ^b^	65.6 ± 4.0 ^c^	60.1 ± 0.9 ^c^	68.9 ± 2.6 A	NS	<0.0001	<0.0001
total flavonols	3.8 ± 0.1 ^a^	3.7 ± 0.1 ^a^	3.0 ± 0.1 ^a^	3.2 ± 0.1 ^a^	3.4 ± 0.1 A	1.9 ± 0.1 ^c^	2.8 ± 0.1 ^b^	3.1 ± 0.1 ^a^	1.7 ± 0.1 ^c^	2.4 ± 0.2 B	<0.0001	<0.0001	<0.0001
quercetin-3-*O*-rutinoside	2.8 ± 0.1 ^a^	2.6 ± 0.1 ^a^	2.0 ± 0.1 ^b^	1.9 ± 0.0 ^b^	2.3 ± 0.1 A	1.0 ± 0.1 ^d^	1.8 ± 0.1 ^b^	1.5 ± 0.0 ^c^	0.7 ± 0.1 ^d^	1.3 ± 0.1 B	<0.0001	<0.0001	<0.0001
quercetin-3-*O*-glucoside	0.2 ± 0.0 ^a^	0.1 ± 0.0 ^b^	0.2 ± 0.0 ^a^	0.2 ± 0.0 ^a^	0.2 ± 0.0 A	0.1 ± 0.0 ^b^	0.1 ± 0.0 ^b^	0.2 ± 0.0 ^a^	0.1 ± 0.0 ^b^	0.1 ± 0.0 B	<0.0001	0.0026	0.0003
kaempferol-3-*O*-glucoside	0.6 ± 0.0 ^c^	0.6 ± 0.0 ^c^	0.5 ± 0.0 ^d^	0.7 ± 0.0 ^b^	0.6 ± 0.0 B	0.6 ± 0.1 ^c^	0.6 ± 0.0 ^c^	1.0 ± 0.0 ^a^	0.5 ± 0.0 ^d^	0.7 ± 0.1 B	0.0031	0.0001	<0.0001
myricetin	0.31 ± 0.01 ^a^	0.29 ± 0.01 ^a^	0.30 ± 0.01 ^a^	0.46 ± 0.01 ^a^	0.34 ± 0.01 A	0.25 ± 0.01 ^a^	0.31 ± 0.01 ^a^	0.38 ± 0.04 ^a^	0.41 ± 0.03 ^a^	0.33 ± 0.01 A	NS	0.0013	NS
total anthocyanins	77.2 ± 1.1 ^b^	110.4 ± 1.1 ^a^	35.7 ± 1.1 ^d^	38.1 ± 8.9 ^d^	65.3 ± 0.7 A	67.5 ± 1.8 ^c^	77.8 ± 2.9 ^b^	62.5 ± 3.9 ^c^	58.4 ± 0.8 ^c^	66.5 ± 2.5 A	NS	<0.0001	<0.0001
cyanidin-3-*O*-rutinoside	71.7 ± 1.1 ^b^	102.7 ± 1.0 ^a^	31.5 ± 1.0 ^d^	33.9 ± 8.5 ^d^	59.9 ± 0.6 A	62.5 ± 1.7 ^c^	72.0 ± 2.7 ^b^	58.7 ± 3.7 ^c^	54.6 ± 0.7 ^c^	61.9 ± 2.2 A	NS	<0.0001	<0.0001
cyanidin-3-*O*-glucoside	5.5 ± 0.1 ^b^	7.7 ± 0.1 ^a^	4.2 ± 0.2 ^c^	4.2 ± 0.4 ^c^	5.4 ± 0.1 A	5.0 ± 0.2 ^b^	5.7 ± 0.2 ^b^	3.9 ± 0.2 ^d^	3.8 ± 0.1 ^d^	4.6 ± 0.2 B	<0.0001	<0.0001	0.0007

Data are presented as the mean ± SE with ANOVA *p*-value; Means in rows followed by the same letter are not significantly different at the 5% level of probability (*p* < 0.05); NS not significant statistically.

**Table 2 molecules-25-03729-t002:** The content of dry matter (in g/100 g FW) and polyphenols (in mg/100 g FW) in examined sour cherry cultivars in 2016.

Bioactive Compounds/Experimental Combination	Organic Sour Cherry (*n* = 12)					Conventional Sour Cherry (*n* = 12)					*p*-Value		
	Keleris 16 cv. (*n* = 3)	Olbacińska cv. (*n* = 3)	Pandy 103 cv. (*n* = 3)	Debreceni Botermo cv. (*n* = 3)	Mean for Organic	Keleris 16 cv. (*n* = 3)	Olbacińska cv. (*n* = 3)	Pandy 103 cv. (*n* = 3)	Debreceni Botermo cv. (*n* = 3)	Mean for Conventional	System (S)	Cultivar (C)	Interaction (SxC)
dry matter	14.9 ± 0.2 ^cd^	16.8 ± 0.6 ^a^	16.7 ± 1.0 ^a^	16.4 ± 0.4 ^a^	16.2 ± 0.4 A	15.4 ± 0.3 ^b^	14.3 ± 0.2 ^d^	14.4 ± 0.9 ^d^	15.1 ± 0.3 ^b^	14.8 ± 0.3 B	0.011	NS	0.0158
total polyphenols	66.2 ± 0.7 ^b^	70.4 ± 1.8 ^ab^	71.2 ± 0.8 ^a^	79.5 ± 1.0 ^a^	71.8 ± 1.5 A	73.2 ± 2.4 ^a^	61.2 ± 2.8 ^b^	56.8 ± 3.1 ^c^	59.1 ± 1.5 ^c^	62.6 ± 2.2 B	<0.0001	NS	<0.0001
total phenolic acids	3.3 ± 0.1 ^a^	3.2 ± 0.0 ^a^	4.1 ± 0.8 ^a^	7.6 ± 0.6 ^a^	4.6 ± 0.6 A	7.1 ± 0.5 ^a^	4.4 ± 1.4 ^a^	2.4 ± 0.1 ^a^	2.2 ± 0.1 ^a^	4.0 ± 0.7 A	NS	NS	NS
ferulic acid	1.7 ± 0.0 ^a^	1.7 ± 0.0 ^a^	1.6 ± 0.1 ^a^	1.5 ± 0.0 ^b^	1.6 ± 0.0 A	1.2 ± 0.0 ^d^	1.4 ± 0.1 ^c^	1.5 ± 0.0 ^b^	1.4 ± 0.0 ^c^	1.4 ± 0.0 B	0.0002	NS	<0.0001
chlorogenic acid	1.6 ± 0.1 ^a^	1.5 ± 0.0 ^a^	2.5 ± 0.9 ^a^	6.1 ± 0.6 ^a^	2.9 ± 0.6 A	5.8 ± 0.5 ^a^	2.9 ± 1.5 ^a^	1.0 ± 0.1 ^a^	0.8 ± 0.0 ^a^	2.6 ± 0.7 A	NS	NS	NS
total flavonoids	62.9 ± 0.6 ^b^	67.2 ± 1.8 ^b^	67.1 ± 1.4 ^b^	71.9 ± 0.4 ^a^	67.3 ± 1.1 A	66.1 ± 2.0 ^b^	56.8 ± 1.6 ^c^	54.3 ± 3.1 ^c^	56.9 ± 1.4 ^c^	58.5 ± 1.7 B	<0.0001	NS	<0.0001
total flavonols	4.6 ± 0.2 ^a^	3.9 ± 0.4 ^b^	3.6 ± 0.1 ^b^	4.3 ± 0.2 ^a^	4.1 ± 0.2 A	4.3 ± 0.1 ^a^	3.0 ± 0.4 ^c^	2.9 ± 0.5 ^c^	3.4 ± 0.1 ^b^	3.4 ± 0.2 B	0.015	0.018	<0.0001
quercetin-3-*O*-rutinoside	1.7 ± 0.1 ^b^	1.5 ± 0.1 ^c^	1.7 ± 0.2 ^b^	2.7 ± 0.2 ^a^	1.9 ± 0.2 A	2.4 ± 0.1 ^a^	1.2 ± 0.3 ^c^	0.8 ± 0.1 ^d^	0.8 ± 0.1 ^d^	1.3 ± 0.2 B	0.0011	0.0084	0.005
quercetin-3-*O*-glucoside	0.7 ± 0.1 ^a^	0.6 ± 0.1 ^a^	0.6 ± 0.0 ^a^	0.6 ± 0.0 ^a^	0.6 ± 0.0 A	0.5 ± 0.0 ^b^	0.4 ± 0.1 ^b^	0.4 ± 0.1 ^b^	0.6 ± 0.0 ^a^	0.5 ± 0.0 B	0.032	NS	<0.0001
kaempferol-3-*O*-glucoside	1.8 ± 0.1 ^a^	1.4 ± 0.2 ^a^	0.9 ± 0.1 ^c^	0.7 ± 0.0 ^c^	1.2 ± 0.1 A	0.6 ± 0.0 ^c^	1.0 ± 0.1 ^bc^	1.3 ± 0.3 ^b^	1.7 ± 0.0 ^a^	1.2 ± 0.1 A	NS	NS	<0.0001
myricetin	0.34 ± 0.03 ^c^	0.48 ± 0.04 ^b^	0.36 ± 0.03 ^c^	0.81 ± 0.07 ^a^	0.49 ± 0.01 A	0.32 ± 0.04 ^c^	0.34 ± 0.03 ^c^	0.34 ± 0.04 ^c^	0.31 ± 0.01 ^d^	0.33 ± 0.03 B	0.044	0.0098	<0.0001
total anthocyanins	58.3 ± 0.5 ^a^	63.3 ± 2.0 ^a^	63.5 ± 1.4 ^a^	67.6 ± 0.6 ^a^	63.2 ± 1.1 A	61.7 ± 1.9 ^a^	53.8 ± 1.2 ^a^	51.5 ± 2.6 ^a^	53.5 ± 1.3 ^a^	55.1 ± 1.5 B	<0.0001	NS	NS
cyanidin-3-*O*-rutinoside	52.6 ± 0.4 ^a^	56.9 ± 1.7 ^a^	57.1 ± 1.0 ^a^	61.6 ± 0.5 ^a^	57.1 ± 1.1 A	55.7 ± 1.8 ^a^	48.4 ± 1.1 ^a^	46.2 ± 2.3 ^a^	48.0 ± 1.2 ^a^	49.6 ± 1.3 B	<0.0001	NS	NS
cyanidin-3-*O*-glucoside	5.7 ± 0.1 ^a^	6.4 ± 0.2 ^a^	6.3 ± 0.4 ^a^	6.0 ± 0.1 ^a^	6.1 ± 0.1 A	6.0 ± 0.1 ^a^	5.4 ± 0.1 ^a^	5.3 ± 0.3 ^a^	5.5 ± 0.1 ^a^	5.6 ± 0.1 B	0.013	NS	NS

Data are presented as the mean ± SE with ANOVA *p*-value; Means in rows followed by the same letter are not significantly different at the 5% level of probability (*p* < 0.05); NS not significant statistically.

**Table 3 molecules-25-03729-t003:** The content of dry matter (in g/100 g FW) and polyphenols (in mg/100 g FW) in examined sour cherry cultivars in 2017.

Bioactive Compounds/Experimental Combination	Organic Sour Cherry (*n* = 12)					Conventional Sour Cherry (*n* = 12)					*p*-Value		
	Keleris 16 cv. (*n* = 3)	Olbacińska cv. (*n* = 3)	Pandy 103 cv. (*n* = 3)	Debreceni Botermo cv. (*n* = 3)	Mean for Organic	Keleris 16 cv.(*n* = 3)	Olbacińska cv. (*n* = 3)	Pandy 103 cv. (*n* = 3)	Debreceni Botermo cv. (*n* = 3)	Mean for Conventional	System (S)	Cultivar (C)	Interaction (SxC)
dry matter	12.8 ± 0.5 ^a^	14.1 ± 0.7 ^a^	13.4 ± 0.4 ^a^	12.4 ± 0.1 ^a^	13.2 ± 0.3 B	12.8 ± 0.1 ^a^	15.7 ± 0.4 ^a^	13.9 ± 0.1 ^a^	14.3 ± 0.3 ^a^	14.2 ± 0.3 A	0.0087	0.0037	NS
total polyphenols	78.4 ± 3.2 ^a^	98.3 ± 3.5 ^a^	38.3 ± 1.0 ^a^	40.5 ± 4.8 ^a^	63.9 ± 7.6 A	66.0 ± 2.7 ^a^	83.4 ± 4.8 ^a^	38.1 ± 0.5 ^a^	37.3 ± 5.8 ^a^	56.2 ± 6.0 B	0.029	<0.0001	NS
total phenolic acids	5.9 ± 0.3 ^a^	4.1 ± 0.1 ^a^	4.4 ± 0.1 ^a^	2.6 ± 0.1 ^c^	4.3 ± 0.4 A	4.4 ± 0.1 ^a^	3.5 ± 0.0 ^b^	3.1 ± 0.2 ^b^	1.7 ± 0.1 ^c^	3.2 ± 0.3 B	<0.0001	<0.0001	0.003
ferulic acid	2.2 ± 0.1 ^a^	0.8 ± 0.0 ^b^	1.0 ± 0.0 ^b^	0.6 ± 0.0 ^c^	1.1 ± 0.2 A	0.6 ± 0.0 ^c^	0.7 ± 0.0 ^b^	0.9 ± 0.0 ^b^	0.5 ± 0.0 ^c^	0.7 ± 0.0 A	<0.0001	<0.0001	<0.0001
chlorogenic acid	3.7 ± 0.2 ^a^	3.3 ± 0.1 ^b^	3.4 ± 0.1 ^b^	2.0 ± 0.1 ^a^	3.1 ± 0.2 A	3.8 ± 0.1 ^a^	2.8 ± 0.1 ^c^	2.1 ± 0.1 ^c^	1.2 ± 0.1 ^d^	2.5 ± 0.3 A	<0.0001	<0.0001	0.0013
total flavonoids	72.5 ± 2.9 ^a^	94.2 ± 3.4 ^a^	33.8 ± 1.0 ^a^	38.0 ± 4.8 ^c^	59.7 ± 7.4 A	61.6 ± 2.7 ^a^	79.8 ± 4.8 ^a^	35.0 ± 0.4 ^a^	35.6 ± 5.8 ^a^	53.0 ± 5.8 A	NS	<0.0001	NS
total flavonols	5.1 ± 0.2 ^a^	2.9 ± 0.1 ^b^	2.5 ± 0.0 ^bc^	2.5 ± 0.1 ^bc^	3.2 ± 0.3 A	2.0 ± 0.1 c	3.1 ± 0.1 b	5.4 ± 0.2 a	2.5 ± 0.0 ^bc^	3.2 ± 0.4 A	NS	<0.0001	<0.0001
quercetin-3-*O*-rutinoside	4.5 ± 0.1 ^a^	2.4 ± 0.1 ^b^	2.0 ± 0.0 ^b^	1.7 ± 0.1 ^c^	2.7 ± 0.3 A	1.0 ± 0.1 ^d^	2.2 ± 0.1 ^b^	1.6 ± 0.1 ^c^	0.8 ± 0.1 ^d^	1.4 ± 0.2 A	<0.0001	<0.0001	<0.0001
quercetin-3-*O*-glucoside	0.1 ± 0.0 ^b^	0.1 ± 0.0 ^b^	0.1 ± 0.0 ^b^	0.1 ± 0.0 ^b^	0.1 ± 0.0 B	0.2 ± 0.0 ^a^	0.1 ± 0.0 ^b^	0.2 ± 0.0 ^a^	0.1 ± 0.0 ^b^	0.2 ± 0.0 A	<0.0001	<0.0001	0.0002
kaempferol-3-*O*-glucoside	0.2 ± 0.0 ^c^	0.1 ± 0.0 ^c^	0.1 ± 0.0 ^c^	0.4 ± 0.0 ^b^	0.2 ± 0.0 B	0.5 ± 0.0 ^b^	0.5 ± 0.0 ^b^	2.5 ± 0.1 ^a^	0.4 ± 0.0 ^b^	1.0 ± 0.2 A	<0.0001	<0.0001	<0.0001
myricetin	0.32 ± 0.01 ^b^	0.28 ± 0.01 ^b^	0.31 ± 0.01 ^b^	0.23 ± 0.02 ^b^	0.28 ± 0.01 B	0.32 ± 0.01 ^b^	0.28 ± 0.01 ^b^	1.14 ± 0.03 ^a^	1.19 ± 0.06 ^a^	0.73 ± 0.13 A	<0.0001	<0.0001	<0.0001
total anthocyanins	67.4 ± 2.7 ^a^	91.3 ± 3.3 ^a^	31.3 ± 0.9 ^a^	35.5 ± 4.9 ^a^	56.4 ± 7.3 A	59.6 ± 2.7 ^a^	76.8 ± 4.6 ^a^	29.6 ± 0.3 ^a^	33.1 ± 5.8 ^a^	49.8 ± 5.9 A	NS	<0.0001	NS
cyanidin-3-*O*-rutinoside	63.3 ± 2.5 ^a^	86.1 ± 3.1 ^a^	28.0 ± 0.9 ^a^	32.4 ± 4.9 ^a^	52.4 ± 7.0 A	55.3 ± 2.7 ^a^	71.6 ± 4.5 ^a^	26.3 ± 0.3 ^a^	29.7 ± 5.7 ^a^	45.7 ± 5.7 A	0.044	<0.0001	NS
cyanidin-3-*O*-glucoside	4.2 ± 0.2 ^a^	5.3 ± 0.3 ^a^	3.3 ± 0.1 ^a^	3.1 ± 0.1 ^a^	4.0 ± 0.3 A	4.3 ± 0.2 ^a^	5.1 ± 0.2 ^a^	3.3 ± 0.0 ^a^	3.4 ± 0.1 ^a^	4.0 ± 0.2 A	NS	<0.0001	NS

Data are presented as the mean ± SE with ANOVA *p*-value; Means in rows followed by the same letter are not significantly different at the 5% level of probability (*p* < 0.05); NS not significant statistically.

**Table 4 molecules-25-03729-t004:** The content of dry matter (in g/100 g FW) and polyphenols (in mg/100 g FW) in examined sour cherry cultivars in 2018.

Bioactive Compounds/Experimental Combination	Organic Sour Cherry (*n* = 12)					Conventional Sour Cherry (*n* = 12)					*p*-Value		
	Keleris 16 cv. (*n* = 3)	Olbacińska cv. (*n* = 3)	Pandy 103 cv. (*n* = 3)	Debreceni Botermo cv. (*n* = 3)	Mean for Organic	Keleris 16 cv. (*n* = 3)	Olbacińska cv. (*n* = 3)	Pandy 103 cv. (*n* = 3)	Debreceni Botermo cv. (*n* = 3)	Mean for Conventional	System (S)	Cultivar (C)	Interaction (SxC)
dry matter	14.1 ± 0.4 ^a^	16.3 ± 0.4 ^a^	15.2 ± 0.1 ^a^	14.3 ± 0.1 ^a^	15.0 ± 0.3 A	14.6 ± 0.3 ^a^	15.2 ± 0.4 ^a^	15.5 ± 0.3 ^a^	13.8 ± 0.5 ^a^	14.8 ± 0.3 A	NS	<0.0001	NS
total polyphenols	115.6 ± 5.0 ^a^	53.0 ± 1.3 ^c^	57.2 ± 6.9 ^c^	101.1 ± 0.9 ^a^	81.7 ± 8.1 A	90.1 ± 1.4 ^b^	48.1 ± 0.7 ^c^	46.6 ± 5.8 ^c^	82.2 ± 5.0 ^b^	66.7 ± 6.0 B	0.0007	<0.0001	<0.0001
total phenolic acids	3.8 ± 0.2 ^c^	5.0 ± 0.2 ^a^	2.9 ± 0.1 ^d^	6.0 ± 0.1 ^a^	4.4 ± 0.3 A	3.0 ± 0.1 ^c^	3.1 ± 0.1 ^c^	1.7 ± 0.0 ^e^	4.4 ± 0.1 ^b^	3.0 ± 0.3 B	<0.0001	<0.0001	<0.0001
ferulic acid	0.7 ± 0.1 ^d^	1.1 ± 0.1 ^b^	0.6 ± 0.0 ^d^	2.2 ± 0.0 ^a^	1.1 ± 0.2 A	0.6 ± 0.1 ^d^	0.9 ± 0.0 ^c^	0.5 ± 0.0 ^d^	0.6 ± 0.0 ^d^	0.6 ± 0.0 B	<0.0001	<0.0001	<0.0001
chlorogenic acid	3.1 ± 0.1 ^a^	3.9 ± 0.1 ^a^	2.3 ± 0.0 ^b^	3.8 ± 0.1 ^a^	3.3 ± 0.2 A	2.5 ± 0.0 ^b^	2.2 ± 0.1 ^b^	1.2 ± 0.0 ^c^	3.8 ± 0.1 ^a^	2.4 ± 0.3 B	<0.0001	<0.0001	<0.0001
total flavonoids	111.8 ± 4.8 ^a^	48.0 ± 1.2 ^c^	54.4 ± 6.9 ^c^	95.1 ± 1.0 ^a^	77.3 ± 8.0 A	87.0 ± 1.5 ^ab^	45.0 ± 0.7 ^c^	44.9 ± 5.8 ^c^	77.8 ± 4.9 ^b^	63.7 ± 5.8 B	0.0014	<0.0001	<0.0001
total flavonols	3.3 ± 0.2 ^a^	3.5 ± 0.1 ^a^	3.5 ± 0.1 ^a^	6.5 ± 0.1 ^a^	4.2 ± 0.4 A	3.3 ± 0.0 ^a^	7.2 ± 0.2 ^a^	3.4 ± 0.1 ^a^	2.5 ± 0.1 ^a^	4.1 ± 0.5 A	NS	<0.0001	NS
quercetin-3-*O*-rutinoside	2.7 ± 0.1 ^b^	2.7 ± 0.1 ^b^	2.3 ± 0.1 ^b^	5.7 ± 0.1 ^a^	3.4 ± 0.4 A	2.2 ± 0.1 ^bc^	2.0 ± 0.1 ^c^	0.9 ± 0.1 ^e^	1.2 ± 0.1 ^d^	1.6 ± 0.2 B	<0.0001	<0.0001	<0.0001
quercetin-3-*O*-glucoside	0.1 ± 0.0 ^b^	0.1 ± 0.0 ^b^	0.1 ± 0.0 ^b^	0.1 ± 0.0 ^b^	0.1 ± 0.0 A	0.1 ± 0.0 ^b^	0.2 ± 0.0 ^a^	0.1 ± 0.0 ^b^	0.1 ± 0.0 ^b^	0.1 ± 0.0 A	NS	<0.0001	<0.0001
kaempferol-3-*O*-glucoside	0.1 ± 0.0 ^c^	0.1 ± 0.0 ^c^	0.7 ± 0.0 ^b^	0.2 ± 0.0 ^c^	0.3 ± 0.1 B	0.6 ± 0.1 ^b^	3.3 ± 0.1 ^a^	0.6 ± 0.0 ^b^	0.6 ± 0.1 ^b^	1.3 ± 0.3 A	<0.0001	<0.0001	<0.0001
myricetin	0.48 ± 0.01 ^b^	0.37 ± 0.02 ^b^	0.50 ± 0.02 ^b^	0.37 ± 0.03 ^b^	0.43 ± 0.02 B	0.43 ± 0.04 ^b^	0.35 ± 0.01 ^a^	1.68 ± 0.06 ^a^	1.74 ± 0.03 ^b^	1.05 ± 0.19 A	<0.0001	<0.0001	<0.0001
total anthocyanins	108.4 ± 4.7 ^a^	44.5 ± 1.1 ^e^	50.9 ± 7.0 ^d^	88.6 ± 0.9 ^b^	73.1 ± 7.9 A	83.7 ± 0.7 ^b^	37.9 ± 0.7 ^f^	41.5 ± 5.9 ^e^	75.4 ± 5.9 ^c^	59.6 ± 6.1 B	0.0015	<0.0001	<0.0001
cyanidin-3-*O*-rutinoside	103.1 ± 4.5 ^a^	40.5 ± 1.0 ^c^	47.1 ± 7.0 ^a^	84.0 ± 0.9 ^b^	68.7 ± 7.8 A	79.0 ± 0.7 ^b^	34.3 ± 0.7 ^c^	37.8 ± 5.9 ^c^	70.7 ± 5.9 ^b^	55.4 ± 6.0 B	0.0016	<0.0001	<0.0001
cyanidin-3-*O*-glucoside	5.3 ± 0.2 ^a^	4.0 ± 0.1 ^a^	3.8 ± 0.1 ^a^	4.6 ± 0.1 ^a^	4.4 ± 0.2 A	4.8 ± 0.1 ^a^	3.6 ± 0.1 ^a^	3.7 ± 0.1 ^a^	4.6 ± 0.1 ^a^	4.2 ± 0.2 A	NS	<0.0001	NS

Data are presented as the mean ± SE with ANOVA *p*-value; Means in rows followed by the same letter are not significantly different at the 5% level of probability (*p* < 0.05); NS not significant statistically.

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
