# Peer review of "The Dynamic of Polyphenols Concentrations in Organic and Conventional Sour Cherry Fruits: Results of a 4-Year Field Study"

_molecules, 2020, doi:10.3390/molecules25163729_

Round 1

Reviewer 1 Report

The manuscript "The concentration of polyphenols in organic and conventional sour cherry fruits in time of long-term experiment" is very interesting providing some new insights regarding the content in polyphenols for organic and conventional sour cherry. For a better clarity I suggest a small revision of the title, e.g.: "The dynamic of polyphenols concentrations in organic and conventional cherry fruits in a long-term experiment" or something similar.

Page 1, row: What does "k." means?

Page 2, line 52-53: "but only in the first (p=0.0078) and second (p=0.011) experimental years." - According to the tables there were 4 experimental years, but in abstract "two-year experiment" is mentioned, which at a first reading creates confusion.  A more clear phrase articulation is needed in the abstract: e.g. "Given that only one two-year experiment on the status of polyphenols in sour cherry fruits in an organic farm was previously conducted and found in the existing scientific literature, the aim of this ..... "

One of the major concerns is the one regarding the tables. The tables 1-4 headings are not very clear and somehow confusing. The values from the column "organic sour cherry" are the mean values obtained from all cultivars grown in organic system? The same question goes for "conventional sour cherry" column? If yes, why? Which is the relevance of comparing these mean values? In the columns with the cultivars, the values are for organic or for conventional sour cherries samples? A revision of the tables is mandatory.

Also, there are some type-o/spelling errors and I recommend a careful revision of the manuscript English style and grammar. 

Why on the footer of some pages appears Antioxidants journals while on other appears Molecules journal?

Author Response

Reviewer 1

Comment 1: The manuscript "The concentration of polyphenols in organic and conventional sour cherry fruits in time of long-term experiment" is very interesting providing some new insights regarding the content in polyphenols for organic and conventional sour cherry. For a better clarity I suggest a small revision of the title, e.g.: "The dynamic of polyphenols concentrations in organic and conventional cherry fruits in a long-term experiment" or something similar.

Authors’ response: Thank you for pointing of title changing. Authors agree with Reviewer comment. The manuscript title was changed into more adequate.

Comment 2: Page 1, row: What does "k." means?

Authors’ response:  Authors want to apologise for mistake. It is an Authors’ mistake. Should be word “thousand”.  The pointed error was corrected into manuscript text.

Comment 3: Page 2, line 52-53: "but only in the first (p=0.0078) and second (p=0.011) experimental years." - According to the tables there were 4 experimental years, but in abstract "two-year experiment" is mentioned, which at a first reading creates confusion.  A more clear phrase articulation is needed in the abstract: e.g. "Given that only one two-year experiment on the status of polyphenols in sour cherry fruits in an organic farm was previously conducted and found in the existing scientific literature, the aim of this ..... "

Authors’ response:  Authors want to apologise. It was that was an unfortunate term. The actually presented experiment was carried out on a 4-year experiment basis. In the abstract it is mentioned that there is only one experiment available in the literature describing the content of polyphenols in organic fruit and conventional sour cherries during the two years of the experiment. In the abstract, the wrongly worded sentence has been corrected as suggested by the Reviewer.

Comment 4: One of the major concerns is the one regarding the tables. The tables 1-4 headings are not very clear and somehow confusing. The values from the column "organic sour cherry" are the mean values obtained from all cultivars grown in organic system? The same question goes for "conventional sour cherry" column? If yes, why? Which is the relevance of comparing these mean values? In the columns with the cultivars, the values are for organic or for conventional sour cherries samples? A revision of the tables is mandatory.

Authors’ response:  Authors agree with Reviewer comment. Tables 1-4 are completely revision. Table content show now individual results for all examined cultivars in both systems (organic and conventional)

Comment 5: Also, there are some type-o/spelling errors and I recommend a careful revision of the manuscript English style and grammar.

Authors’ response:  Manuscript was edited for proper English language, grammar, punctuation, spelling, and overall style by one or more of the highly qualified native English speaking editors at AJE.

Comment 6: Why on the footer of some pages appears Antioxidants journals while on other appears Molecules journal?

Authors’ response:  Authors’ want to apologise. First time manuscript was submitted to Antioxidants journal. After two days Editor gave us negative decision about processing manuscript. So Authors decided to send it to another journal Molecules. Because Molecules belong to the same group of journal (MDPI) authors use the same type of template, but forgot change name of journal in footnote.  The error was corrected in manuscript text.

Reviewer 2 Report

Molecules- 889299

Revision of the titled manuscript “The concentration of polyphenols in organic and conventional sour cherry fruits in time of long-term  experiment

The manuscript under revision compare the concentration of bioactive compounds in organic and conventional sour cherries and to determine the effects of cultivation year and the proper cultivar. However, it does not make a very adequate description of the results and their argumentation is lacking.

The manuscript cannot be accepted in the present form. Major revision is proposed.

There are several aspects that can be improved:

In the title:

The expresión “ in time of long-term experiment” is not clear. It is not understood that the effect of time on the implantation of organic culture should be evaluated.

In the abstract section:

If the analysis of the fruits has been carried out for 4 years (2015-2018), it does not make sense that lines 14-15 indicate that:”Given that only one two-year experiment on the status of polyphenols …”

In  Material and Methods section:

2.4. Polyphenol extraction and identification    (line 198) “method19” should be changed.

The analysis of the results is a simply description of the results. Some explanation for why the levels of phenolic compounds increase must be argued.

 In the Results and discusión section:

Also some aspects must be taken into account:

Line 55: “Fruits produced under organic conditions contained more dry matter” should be revised, indicating that some authors have demonstrated it, etc ... but it remains incomplete.

Lines 56-57: “The increased level of dry matter content in organic fruits can be explained by the water swelling” phenomena.” In this sentence it must be specified that the phenomenon occurs in conventional fruits.

Lines 86-87: “Among the examined cultivars, ‘Oblacinska’ cv. exhibited the highest levels of total polyphenols.” This statement is not correct, there are only significant differences regarding the content of total polyphenols in 2015 and 2017.

Tables 1-4: It is not understood how the results are expressed. The first two columns correspond to the average of all organic and conventional cultivars, but the following columns that represent (cv) is only the value of the conventional cultivar one. If in the work the authors want to see the evolution of the organic ones because only the conventional ones are presented?  This aspect must be clarified.

The statistic analysis should be reviewed in the following data:

Total flavonoids  “organic sour cherry” and “conventional sour cherry” in 2015

Myricetin “organic sour cherry” and “conventional sour cherry” in 2015

Author Response

Reviewer 2

Revision of the titled manuscript “The concentration of polyphenols in organic and conventional sour cherry fruits in time of long-term  experiment”

The manuscript under revision compare the concentration of bioactive compounds in organic and conventional sour cherries and to determine the effects of cultivation year and the proper cultivar. However, it does not make a very adequate description of the results and their argumentation is lacking.

The manuscript cannot be accepted in the present form. Major revision is proposed.

There are several aspects that can be improved:

Comment 1: In the title: The expresión “ in time of long-term experiment” is not clear. It is not understood that the effect of time on the implantation of organic culture should be evaluated.

Authors’ response:  Authors’ agree with Reviewer comment. According to Reviewer 1 and 2 suggestion the title of manuscript is change into: “The dynamic of polyphenols concentrations in organic and conventional sour cherry fruits: Results of a 4-year field study”

Comment 2: In the abstract section: If the analysis of the fruits has been carried out for 4 years (2015-2018), it does not make sense that lines 14-15 indicate that:” Given that only one two-year experiment on the status of polyphenols …”

Authors’ response:  Similar remark was pointed by Reviewer no. 1. The incorrectly written sentence has been corrected in the manuscript Abstract.

Comment 3: In  Material and Methods section:

2.4. Polyphenol extraction and identification    (line 198) “method19” should be changed.

Authors’ response:  The reference number 19 has been corrected and placed in square brackets [19]

Comment 4: The analysis of the results is a simply description of the results. Some explanation for why the levels of phenolic compounds increase must be argued.

Authors’ response:  According to Reviewer suggestion some new elements of discussion are added into manuscript text. Authors give explanation of different levels of individual phenolic acids and flavonols in sour cherry fruits.

Comment 5: In the Results and discusión section:

Also some aspects must be taken into account:

Line 55: “Fruits produced under organic conditions contained more dry matter” should be revised, indicating that some authors have demonstrated it, etc ... but it remains incomplete.

Authors’ response:  According to Reviewer suggestion sentence was corrected

Comment 6: Lines 56-57: “The increased level of dry matter content in organic fruits can be explained by the water swelling” phenomena.” In this sentence it must be specified that the phenomenon occurs in conventional fruits.

Authors’ response:  According to Reviewer suggestion sentence was corrected

Comment 7: Lines 86-87: “Among the examined cultivars, ‘Oblacinska’ cv. exhibited the highest levels of total polyphenols.” This statement is not correct, there are only significant differences regarding the content of total polyphenols in 2015 and 2017.

Authors’ response:  According to Reviewer suggestion all data connected with the total polyphenols content was carefully checked and corrected into manuscript text.

Comment 8: Tables 1-4: It is not understood how the results are expressed. The first two columns correspond to the average of all organic and conventional cultivars, but the following columns that represent (cv) is only the value of the conventional cultivar one. If in the work the authors want to see the evolution of the organic ones because only the conventional ones are presented?  This aspect must be clarified.

Authors’ response:  In accordance to suggestion of Reviewer no. 1 and 2 all tables were completely revision. Data presented in the New Tables show individual results for examined cultivars separately in organic and conventional systems.

Comment 9: The statistic analysis should be reviewed in the following data:

Total flavonoids  “organic sour cherry” and “conventional sour cherry” in 2015

Authors’ response:  Authors want to apologise. Of course it was a mistake. Both values for total flavonoids content in 2015: 68.8±0.7 (organic sour cherry) and 68.9±2.6 (conventional sour cherry) are not significant statistically and should be labelled by the same letter (A). A mistake is corrected into Table 1.

Comment 10: Myricetin “organic sour cherry” and “conventional sour cherry” in 2015

Authors’ response: Authors want to apologise. Of course it was a mistake. Both values for myricetin content in 2015: 0.3±0.0 (organic sour cherry) and 0.3±0.0 (conventional sour cherry) are not significant statistically and should be labelled by the same letter (A). A mistake is corrected into Table 1.

Reviewer 3 Report

General

The results presented in the manuscript compare phytochemical composition of sour cherries cultivated by organic and conventional agro-technologies. Many studies have been performed on this topic with different crops and the results obtained so far are rather controversial, i.e. some of them have not found any significant differences, while the others reported higher amounts of some bioactive compounds in organic crops. Therefore, the research on this topic remains interesting and important, particularly due to the increasing popularity of organic fruits and vegetables.

However, in the current state the manuscript seems rather preliminary and inconsistently written; therefore, it needs major revision. First of all, Results and discussion section should be completely rewritten. The manuscript contains a lot of data, some results are rather controversial (Tables 1-4), while in the current state this section consists of the purposefully selected, very short and in most cases highly simplified statements about the measured values (‘higher’, ‘lower’, ‘increased’, etc.), which are presented in the tables. Several other important factors may play an important role in the variations of these numbers and they are not discussed. Some selected remarks are provided below; however, most of them can be applied to the whole section.

Sampling procedure is missing in the Materials and methods section, while it is an extremely important issue for this kind of experimental work. The variations in the measured values may highly depend on the sample representativeness (the amount of the total cherries included into each sample, harvesting scheme from each tree, the number of trees for the sample, etc.). In general, Materials and methods section is lacking many important details, which would be required for repeating the procedures independently.

Some studies reported that organic crops may accumulate higher amounts of secondary metabolites; however, their total yields in conventional agriculture may be remarkably higher compared to the organic farming. If the total yields have been determined, they should be included. In this case it would be possible to calculated how much of important bioactives may be obtained from the determined area of land.

Language quality should be improved. I have made some remarks but further editing is required.

Specific remarks

Line 24: So far as the first 2 keywords are already present in the title I suggest replacing them by other keywords, which are important for this study; e.g. instead of ‘sour cherry’ to give its botanical name.

Line 28: ‘after’? The numbers show that ‘followed by’.

Lies 32-33: What difference between fresh fruits and dessert fruits?

Line 35: Why management? Organic farming itself is a strict system.

Line 36: Why ‘natural methods of fertilization’? Natural fertilizers are listed in the brackets.

Line 38: ‘ideal’ is definitely too strong word. There are many other fruits and vegetables, which are excellent and even better sources of bioactive compounds.

Line 40: ‘stressors’ = stress

Line 40: Why ‘on the other hand’?

Line 43: In the meantime it is just hypothesized that they ‘can prevent”; sound and unambiguous proves have not been obtained yet.

Line 50: ‘after’ = during

Lines 56-60: Conventional fruits had higher dry matter values in 2017, while in 2018 the differences were not significant. Consequently, this explanation seems rather speculative and valid only for some fraction of the results obtained. If the reason is ‘water swelling’ phenomena, why sometimes it is not valid?

Tables 1-4: The layout of the values should explained more clearly. I understand that the values for organic and conventional samples were calculated for all cultivars/samples (4x3 replicates =12), while the means for individual cultivars are calculated both from organic and conventional samples (2 x 3 replicates = 6). Was it reasonable to mix organic and conventional samples in the latter case? Why not to give the results separately for each cultivar? May be, some interesting and specific variations could be revealed in this case for each cultivar?

Table 1: 74.07±9015? Most likely it is a mistake. Is it correct that 68.93±2.55 is smaller than 68.76±8.9 for total flavonoids in organic and conventional fruits?

Lines 73-160: All text is written as one paragraph. In general, results and discussion section should be better organized by grouping the results and discussing them in the separated subsections.

Lines 84-86: These results indicated that the method of farming is not the only factor affecting the accumulation of phenolic compounds. Harvesting time, climatic conditions may be other important factors. To my opinion, the authors should include these factors in their discussion. Moreover, they have climate data for each year.

Line 102-104: This is rather speculative explanation. It may be understood that ‘experimental year’ is the most important factor for flavonoids. E.g. if you start experiment in 2018, the content will increase in 2019, 2020, etc. Surely, that other reasons are responsible for these variations, e.g. harvesting time, climatic conditions, may be, even the age of the tree? The discussion should be expanded and more consistent. Currently it consists mainly of the bare and simple statements and numbers.

Line 110: should be myricetin

Line 124: should be cyanidin

Lines 124-125: It seems that the values are discussed selectively, e.g. by emphasizing the superiority of organic fruits. However, the content of kaempferol-3-O-rutinoside and myricetin was several times higher in conventionally grown cherries (Tables 3, 4).How to explain?

Line 198: ‘method19’?

Line 199: It is not clear from what kind of material these 100 mg were taken. How freeze-dried sample was prepared before weighing? It is very important for the accuracy of HPLC analysis.

Line 202: What means 0.9 mL of extract? When 1 mL of methanol was used it should be all (or almost all) liquid extract.

Line 212: ‘(Poznan, Poland)’ should be deleted.

Line 216: If 1 mL of solvent was used for extraction (section 2.4) how to collect 2.5 mL of supernatant?

Lines 221-222: Should be cyanidin

Line 222: To the best of my knowledge this purity anthocyanin standards are not available from Sigma-Aldrich.

Section 2.6: It is not indicated how many replicate extractions and HPLC runs were performed in the analysis of polyphenols.

Lines 226-228: The number and origin of the samples should be explained more clearly: what means n=12 when 4 different cultivars were studied? 3 samples for each cultivar for organic and conventional farming?

Author Response

Reviewer 3

The results presented in the manuscript compare phytochemical composition of sour cherries cultivated by organic and conventional agro-technologies. Many studies have been performed on this topic with different crops and the results obtained so far are rather controversial, i.e. some of them have not found any significant differences, while the others reported higher amounts of some bioactive compounds in organic crops. Therefore, the research on this topic remains interesting and important, particularly due to the increasing popularity of organic fruits and vegetables.

However, in the current state the manuscript seems rather preliminary and inconsistently written; therefore, it needs major revision.

Comment 1: First of all, Results and discussion section should be completely rewritten. The manuscript contains a lot of data, some results are rather controversial (Tables 1-4), while in the current state this section consists of the purposefully selected, very short and in most cases highly simplified statements about the measured values (‘higher’, ‘lower’, ‘increased’, etc.), which are presented in the tables. Several other important factors may play an important role in the variations of these numbers and they are not discussed. Some selected remarks are provided below; however, most of them can be applied to the whole section.

Authors’ response: Base on all previous Reviewers comments and suggestion as well using elements pointed by Reviewer no. 3 the content of the tables 1-4 was completely revision. If the Authors use in manuscript text statements as: “higher”, “lower” so the statements are focused by results of statistical analysis. If the authors use “increased” sentence, so the trends is as well reflected by the values in Tables among the experimental years. Tables content show now individual results for all examined cultivars in both systems (organic and conventional). This are data which were not show in the first version of manuscript.

Comment 2: Sampling procedure is missing in the Materials and methods section, while it is an extremely important issue for this kind of experimental work. The variations in the measured values may highly depend on the sample representativeness (the amount of the total cherries included into each sample, harvesting scheme from each tree, the number of trees for the sample, etc.). In general, Materials and methods section is lacking many important details, which would be required for repeating the procedures independently.

Authors’ response: According to Reviewer suggestion all missing data about fruits collecting and samples preparation before analysis was described in the new sub-section M&M:

2.3. Fruits collecting and sample preparation

Comment 3: Some studies reported that organic crops may accumulate higher amounts of secondary metabolites; however, their total yields in conventional agriculture may be remarkably higher compared to the organic farming. If the total yields have been determined, they should be included. In this case it would be possible to calculated how much of important bioactives may be obtained from the determined area of land.

Authors’ response: Authors agree with Reviewer comment. The calculation of the results for the land area it could give another light for differences between organic and conventional fruits production. Authors want to thank you for pointing of the new directions of experiments in the future. In presented manuscript the aim of the experiment was to evaluate the content of bioactive compounds in fruits and results are presented in grams and milligrams per 100 g FW. It is important form consumer point of view and pro-healthy sour cherry properties.

Comment 4: Language quality should be improved. I have made some remarks but further editing is required.

Authors’ response:  Manuscript was edited for proper English language, grammar, punctuation, spelling, and overall style by one or more of the highly qualified native English speaking editors at AJE.

Specific remarks

Comment 5: Line 24: So far as the first 2 keywords are already present in the title I suggest replacing them by other keywords, which are important for this study; e.g. instead of ‘sour cherry’ to give its botanical name.

Authors’ response:  According to Reviewer suggestion first two keywords are changed from: “organic sour cherry”, “conventional sour cherry” into: “Prunus cerasus”, “organic”, “conventional”

Comment 6: Line 28: ‘after’? The numbers show that ‘followed by’.

Authors’ response:  According to Reviewer suggestion word “after” was replaced by “followed by”

Comment 7: Lies 32-33: What difference between fresh fruits and dessert fruits?

Authors’ response:  Authors want to apologise. Of course it was a mistake. There is no difference between fresh and dessert fruits. The sentence was correct in manuscript text.

Comment 8: Line 35: Why management? Organic farming itself is a strict system.

Authors’ response:  Authors agree with Reviewer comment. The sentence was correct into manuscript text

Comment 9: Line 36: Why ‘natural methods of fertilization’? Natural fertilizers are listed in the brackets.

Authors’ response:  Authors agree with Reviewer comment. The sentence was correct into manuscript text

Comment 10: Line 38: ‘ideal’ is definitely too strong word. There are many other fruits and vegetables, which are excellent and even better sources of bioactive compounds.

Authors’ response:  Authors agree with Reviewer comment. Too much strong sound word “ideal” was replaced by word “good” into manuscript text.

Comment 11: Line 40: ‘stressors’ = stress

Authors’ response:  Authors agree with Reviewer comment. The sentence was correct into manuscript text.

Comment 12: Line 40: Why ‘on the other hand’?

Authors’ response:  According to reviewer suggestion sentence was correct into manuscript text.

Comment 13: Line 43: In the meantime it is just hypothesized that they ‘can prevent”; sound and unambiguous proves have not been obtained yet.

Authors’ response:  Authors agree with Reviewer comment. The sentence was correct into manuscript text.

Comment 14: Line 50: ‘after’ = during

Authors’ response:  Authors agree with Reviewer comment. The sentence was correct into manuscript text.

Comment 15: Lines 56-60: Conventional fruits had higher dry matter values in 2017, while in 2018 the differences were not significant. Consequently, this explanation seems rather speculative and valid only for some fraction of the results obtained. If the reason is ‘water swelling’ phenomena, why sometimes it is not valid?

Authors’ response:  Authors want to underline, that in time of first two years of experiment organic sour cherry contained a significant higher content of dry matter. In third year of experiment the situation was change and conventional fruits contained more dry matter. In 2018 (fourth experimental year) we observed some tendency to the higher content of dry matter in organic sour cherry fruits again. In long-term experiment such situation are happened. Authors try to explain why organic sour cherry contain more dry matter on the basis of  available references. That is why we use “water swelling” phenomena. Of course, it seems extremely interesting to carefully examine and make further observations in this area. May be in third year of experiment other (unknown) factors are appeared which interrupt normal metabolism of sour cherry trees in organic orchard. On the other hand Authors are gratitude to Reviewer pointing and in the future experiment they want to observed that phenomena more detailed.

Comment 16: Tables 1-4: The layout of the values should explained more clearly. I understand that the values for organic and conventional samples were calculated for all cultivars/samples (4x3 replicates =12), while the means for individual cultivars are calculated both from organic and conventional samples (2 x 3 replicates = 6). Was it reasonable to mix organic and conventional samples in the latter case? Why not to give the results separately for each cultivar? May be, some interesting and specific variations could be revealed in this case for each cultivar?

Authors’ response:  Authors agree with Reviewer comment. As was explain it before Tables 1-4 were completely revised. In the new tables all interaction between cultivars and cultivation system was presented. It gave a new light into experimental data and the way of their discussion.

Comment 17: Table 1: 74.07±9015? Most likely it is a mistake. Is it correct that 68.93±2.55 is smaller than 68.76±8.9 for total flavonoids in organic and conventional fruits?

Authors’ response:  Authors want to apologise. Because all tables were completely revised, all errors were checked again and corrected.

Comment 18: Lines 73-160: All text is written as one paragraph. In general, results and discussion section should be better organized by grouping the results and discussing them in the separated subsections.

Authors’ response:  Authors agree with Reviewer remark. The two previous Reviewers as well noted this fact. Therefore, the Authors have divided the chapter Results and Discussion into sub-sections. This makes the text much easier to read and understand.

Comment 19: Lines 84-86: These results indicated that the method of farming is not the only factor affecting the accumulation of phenolic compounds. Harvesting time, climatic conditions may be other important factors. To my opinion, the authors should include these factors in their discussion. Moreover, they have climate data for each year.

Authors’ response:  According to Reviewer suggestion some new part of discussion have been added. Especially part with explanation of changing in individual phenolic compounds in fruits. Authors use some references showing, similar problems and explanation with weather condition in time of fruits production. 

Comment 20: Line 102-104: This is rather speculative explanation. It may be understood that ‘experimental year’ is the most important factor for flavonoids. E.g. if you start experiment in 2018, the content will increase in 2019, 2020, etc. Surely, that other reasons are responsible for these variations, e.g. harvesting time, climatic conditions, may be, even the age of the tree? The discussion should be expanded and more consistent. Currently it consists mainly of the bare and simple statements and numbers.

Authors’ response:  The observations of changes in the concentration of total polyphenols were conducted in the next four years of the experiment. Of course, if the experiment  have been carried out in other years, the results could have been different. However, the authors emphasize that both experimental orchards are located very close to each other (16 km). Therefore, the climatic and weather conditions during cultivation are very similar. All other conditions (which can be controlled during cultivation) are also very similar. The age of the trees, the rootstock and the type of soil in the orchard, as well harvest time were also very similar. Only one difference between this two cultivation systems are the treatments used in the organic orchard and others in the conventional orchard. Another factor that differentiates the cultivation years is the weather in the following year. This information is also detailed in the presented manuscript.

Comment 21: Line 110: should be myricetin

Authors’ response:  Authors agree with Reviewer remark. The error was correct in manuscript text.

Comment 22: Line 124: should be cyanidin

Authors’ response:  Authors agree with Reviewer remark. The error was correct in manuscript text.

Comment 23: Lines 124-125: It seems that the values are discussed selectively, e.g. by emphasizing the superiority of organic fruits. However, the content of kaempferol-3-O-rutinoside and myricetin was several times higher in conventionally grown cherries (Tables 3, 4).How to explain?

Authors’ response:  Authors want to apologise. A relevant description and discussion of the obtained kaempferol-3-O-glucoside content results in conventional cherries was added to the discussion sub-section.

Comment 24: Line 198: ‘method19’?

Authors’ response:  it should be: by HPLC method [19]. The sentence was corrected in manuscript text.

Comment 24: Line 199: It is not clear from what kind of material these 100 mg were taken. How freeze-dried sample was prepared before weighing? It is very important for the accuracy of HPLC analysis.

Authors’ response:  Authors want to apologise. A missing information about preparation samples and freeze-dying process was added into manuscript text in sub-section:

2.3. Fruits collecting and sample preparation

Comment 25: Line 202: What means 0.9 mL of extract? When 1 mL of methanol was used it should be all (or almost all) liquid extract.

Authors’ response:  Authors want to apologise. The incorrectly described sample preparation procedure and its extraction steps were corrected in the manuscript (line 240-244)

Comment 26: Line 212: ‘(Poznan, Poland)’ should be deleted.

Authors’ response:  According to Reviewer suggestion name of city and country of chemical standards distributor were deleted.

Comment 27: Line 216: If 1 mL of solvent was used for extraction (section 2.4) how to collect 2.5 mL of supernatant?

Authors’ response:  Authors want to apologise. Not 1 mL but 5 mL of methanol was added to the freeze-dried plant material. Incorrect value was corrected in manuscript section (Line 240)

Comment 28: Lines 221-222: Should be cyanidin

Authors’ response:  According to Reviewer suggestion. Name of incorrect anthocyanin was corrected in manuscript text (Lines 262-263).

Comment 29: Line 222: To the best of my knowledge this purity anthocyanin standards are not available from Sigma-Aldrich.

Authors’ response:  Authors want to apologise for mistake. To more clarified source of phenolic compounds used as an standards, their purchase and purity, CAS numbers was added to all polyphenol compounds.

Comment 30: Section 2.6: It is not indicated how many replicate extractions and HPLC runs were performed in the analysis of polyphenols.

Authors’ response:  Authors want to explain number of HPLC replication. Individual  orchard sample (per tree) was treated as a field replication and this is explained in sub-section 2.7 Statistical Analysis. Additionally, this information is given in the tables. The number of HPLC replicates (laboratory replicates) was n = 3 to obtain stability of the results obtained.

Comment 31: Lines 226-228: The number and origin of the samples should be explained more clearly: what means n=12 when 4 different cultivars were studied? 3 samples for each cultivar for organic and conventional farming?

Authors’ response: Authors want to explain number of all experimental samples and replications. All missing data were describe in sub-section 2.7 Statistical Analysis. One tree was treated as field replication (n=3) per cultivar. Four cultivars were tested by every year in each cultivation system (4 x 3 = 12). The number of laboratory replication (HPLC extraction and injections) n=3.

Round 2

Reviewer 1 Report

The authors took in consideration the reviewer suggestions and made the necessary changes and can be accepted for publishing after a minor revision of English and in text editing. There are still some type-o/spelling errors that should be corrected (e.g. the word "thousand" is written "thousend" - line 33). 

Author Response

Reviewer 1

Comment 1: The authors took in consideration the reviewer suggestions and made the necessary changes and can be accepted for publishing after a minor revision of English and in text editing. There are still some type-o/spelling errors that should be corrected (e.g. the word "thousand" is written "thousend" - line 33).

Authors’ response: Authors want to apologize for pointing of  spelling error. Word “thousand” was corrected into “thousand”

Reviewer 2 Report

I consider that all my comments have been taken into account and therefore the manuscript should be published in this form.

Author Response

Reviewer 2

Comment 1: I consider that all my comments have been taken into account and therefore the manuscript should be published in this form.

Authors’ response: Authors are grateful for positive Reviewer recommendation for publication

Reviewer 3 Report

The manuscript has been properly revised and its quality has been improved. However, it still contains some shortcomings, which have to be eliminated. The authors answered that the manuscript was carefully checked for language quality; however, it still contains quite a few grammar mistakes.

Lines 21-24: To my opinion some conclusive statements in the abstract should be in a better agreement with the results obtained. For instance, the statement that ‚organic samples contained significantly more dry matter’ is contradictory to the results from the year 2017 and 2018. Moreover, when data for individual cultivars are presented for each year, the disagreements with this statement are even more evident (for some cultivars dry matter content was not significantly different in 2015 and 2016). Consequently, such statements are not scientifically sound because they are only partially valid. It is not good idea to select the results, which ‘I like’ and to ignore those which ‘I do not like’. It may be understood that in some cases the results are difficult to explain; however, true findings should be properly discussed in any case.

Line 25: ‘organic and ‘conventional’ are too generic (uninformative) words; should be supplemented by the word ‘farming’

Tables: The results of statistical data handling in some cases seem rather surprising: e.g. there was no significant differences for myricetin measured at the conc. 0.2 and 0.5 in Table 1 (although SD was 0.0); however, in Table 2 0.5 was higher than 0.4, in Table 4 again the difference between 0.5 and 0.4 was absent. Needs checking.

Lines 277-284: The conclusions are too generic. I am not sure that long term experiment (4 years) provided more accurate answers regarding the effect of organic farming. Data variations between years, which would be rather difficult to explain only by the farming system, indicate that other factors might be even more important.

Author Response

Reviewer 3

Comment 1: Lines 21-24: To my opinion some conclusive statements in the abstract should be in a better agreement with the results obtained. For instance, the statement that ‚organic samples contained significantly more dry matter’ is contradictory to the results from the year 2017 and 2018. Moreover, when data for individual cultivars are presented for each year, the disagreements with this statement are even more evident (for some cultivars dry matter content was not significantly different in 2015 and 2016). Consequently, such statements are not scientifically sound because they are only partially valid. It is not good idea to select the results, which ‘I like’ and to ignore those which ‘I do not like’. It may be understood that in some cases the results are difficult to explain; however, true findings should be properly discussed in any case.

Authors’ response:…Authors agree with Reviewer remark. Summary statement was corrected according to data presented in all tables. Because in the group of polyphenolic compounds we can find different (other) compounds as total phenolic acids and  total flavonoids authors precisely pointed in each years the higher concentration of compounds from chemical groups were abundant in organic and in conventional sour cherries.  Authors hope, that now the conclusions presented in sub-section (Summary) are presented without authors preferences. Authors tried to show everything in independence way not as “I like organic” so I presented only better side of obtained results for organic sour cherry.

Comment 2: Line 25: ‘organic and ‘conventional’ are too generic (uninformative) words; should be supplemented by the word ‘farming’

Authors’ response:…Authors want to apologize for using of not formal words in scientific manuscript.  That mistake was corrected in manuscript text.

Comment 3: Tables: The results of statistical data handling in some cases seem rather surprising: e.g. there was no significant differences for myricetin measured at the conc. 0.2 and 0.5 in Table 1 (although SD was 0.0); however, in Table 2 0.5 was higher than 0.4, in Table 4 again the difference between 0.5 and 0.4 was absent. Needs checking.

Authors’ response:… Authors want to clarified pointed by Reviewer problem. Because in the tables use only one decimal position, it could happen that there were no visible differences between the values. As noted by the reviewer, in the case of myricetin, two decimal places were restored to make the differences more visible.

Comment 4: Lines 277-284: The conclusions are too generic. I am not sure that long term experiment (4 years) provided more accurate answers regarding the effect of organic farming. Data variations between years, which would be rather difficult to explain only by the farming system, indicate that other factors might be even more important.

Authors’ response:…Authors want to clarified why they continue more than 2-years long experiment. In organic system the balance between organic soil life, plant condition and the effect of cultivation is taken with years. If Authors obtained a clear results into two first years, probably experiment would be finish after two years. In our case total polyphenols content in the first year of experiment was not significant between systems, second year results was positive for organic system. So after two years we could not answer for question which system is better for sour cherry production. Long term experiment is more preferable.

Similar situation was presented by:

Ren et al. (2017) with two different onion cultivars (6-years experiment);

Mitchell et al. (2007) with tomato (10-years experiment);

Swezey et al (1998) with apple (3-years experiment);

Bryk et al. (2008) with 13 apples cultivars (3-years experiment);

Michereff-Filho et al. (2008) with cabbage (3-years experiment);

Hajšlová et al. (2005) with eight potatoes cultivars (4-years experiment)

Conclusions were changed into much more detailed, to more reflected obtained results.